# Protein and Amino Acid Metabolism in Poultry during and after Heat Stress: A Review

**DOI:** 10.3390/ani11041167

**Published:** 2021-04-19

**Authors:** Mohammed M. Qaid, Maged A. Al-Garadi

**Affiliations:** 1Animal Production Department, College of Food and Agriculture Sciences, King Saud University, Riyadh 11451, Saudi Arabia; 2Veterinary Medicine Faculty, Thamar University, Dhamar 13020, Yemen

**Keywords:** amino acids, broiler, heat stress, heat tolerance, protein metabolism

## Abstract

**Simple Summary:**

Broilers must be reared under thermoneutral conditions and comfort zones; therefore, any deviation from the neutral thermal zone causes stress and a consequent disturbance in the turnover or the metabolism of nutrients. This review addressed the biosynthesis of amino acids and/or protein metabolism under normal conditions and heat stress conditions. In addition, hormonal responses to stress and the role of endocrine hormones in protein metabolism have been reviewed. In addition, the aim of this review is to summarize the studies related to the assessment of heat stress, the physiological stress regulation mechanism, and the nutritional strategies for the prevention of heat stress in poultry.

**Abstract:**

This review examined the influence of environmental heat stress, a concern facing modern broiler producers, on protein metabolism and broiler performance, as well as the physiological mechanisms that activate and control or minimize the detrimental impacts of stress. In addition, available scientific papers that focused on amino acids (AA) digestibility under stress conditions were analyzed. Furthermore, AA supplementation, a good strategy to enhance broiler thermotolerance, amelioration, or stress control, by keeping stress at optimal levels rather than its elimination, plays an important role in the success of poultry breeding. Poultry maintain homeothermy, and their response to heat stress is mainly due to elevated ambient temperature and the failure of effective heat loss, which causes a considerable negative economic impact on the poultry industry worldwide. Reduced feed intake, typically observed during heat stress, was the primary driver for meat production loss. However, accumulating evidence indicates that heat stress influences poultry metabolism and endocrine profiles independently of reduced feed intake. In conclusion, high ambient temperatures significantly reduced dietary AA intake, which in turn reduced protein deposition and growth in broilers. Further studies are required to determine the quantity of the AA needed in warm and hot climates and to introduce genetic tools for animal breeding associated with the heat stress in chickens.

## 1. Introduction

Poultry meat is an essential source of dietary protein, and the industry has developed high grade poultry because of improved farming techniques, automation equipment, and comprehensive and balanced feeding, and other new technologies [1]. In the past decade, broiler production has increased rapidly in tropical and subtropical areas and is expected to sustain robust growth in the future. However, according to a review [2], high environmental temperatures are one of the greatest challenges of poultry and live stock performance, leading to the decline in production efficiency in these countries. Furthermore, modern commercial broilers are more sensitive to heat stress than previous generations due to their higher performance, growth rate, and feed conversion efficiency [3]. Indeed, commercial poultry strains can reach a high production yield, but their body metabolism, being comparatively accelerated, has poor thermoregulation and is poorly adapted to the living environments compared to native backyard chickens [1]. The higher growth rate of broilers has several consequences, such as higher feed consumption and metabolism and elevated production of internal heat. To reduce the heat load and avoid heat-induced mortality in birds, heat loss and/or lowering heat production could be achieved through reduced feed consumption, resulting in a depressed growth rate and lower final body weight, reduced breast meat weight, and lower egg quality, size, and rate in hens. In addition, the decreased feed consumption could result in nutrient deficiency, such as proteins, AA, and energy [4,5,6].

Among the environmental factors, heat stress negatively affects feed consumption, body weight gain [7], and carcass characteristics [2]. Additionally, HS may cause oxidative stress in the body and develop many free radicals, stimulating membrane lipid peroxidation, and hence the degradation of DNA and protein membranes [8,9]. In poultry, mitochondrial superoxide production as oxidative stress was observed on exposure to HS [9]. Besides the role of AA as protein and peptide components, some AA (e.g., glutamine, cysteine, leucine, arginine, tryptophan, and proline) are involved in the regulation of metabolic pathways, thereby affecting growth, protein accumulation, maintenance, immunity, and health [10].

The ambient temperature impacts the protein turnover rate in broiler skeletal muscle [11]. Not only protein anabolism, but also protein catabolism are energetically expensive. The growth depression of heat-exposed chickens showed lower protein gain and retention. High protein sources have beneficial effects through the improved growth of heat-stressed broilers [12]. They indicated that protein synthesis in broilers was more affected than protein breakdown with HS, resulting in reduced protein deposition in the skeletal muscles. A review [11] showed that the high ambient temperatures and dietary protein consumption affected muscle protein turnover in broilers. In broiler chickens, HS alters muscle protein and AA metabolism and accelerates liver gluconeogenesis for energy supply [13]. Dietary approaches, such as modifications of energy and protein content of the diet, are the most practical and preferred ways to alleviate heat distress in poultry and enhance broiler performance under these conditions [14,15]. Improving the overall equilibrium of the dietary AA was more effective than increasing total protein consumption [16].

Limited studies are available that address the effects of HS on protein metabolism in broilers. Therefore, it is necessary to detect mechanisms or methods that allow producers to effectively reduce the detrimental influences of environmental HS on broilers, in particular on protein metabolism via protein accretion or degradation of muscle. The present review will focus on the effects of environmental heat stress on protein metabolism and broiler performance, as well as the physiological mechanisms and nutritional strategies that mitigate the negative effects of heat stress, particularly the role of AA in reducing HS in stressed broilers.

## 2. Amino Acid and/or Protein Metabolism

AA are required for most biological activities. The AA transport into the apical membrane and out of the lateral basal membrane of enterocytes. Their transport relies on sodium-dependent symporters, proton-motive forces, antiporters, and the gradient of other AA. The metabolic fate of absorbed AA mainly depends on nutrient availability [17]. AA moving through catabolic pathways ultimately serve as precursors of gluconeogenesis [13] and contribute to 40% of the total AA loss in fasted animals. Proteins are synthesized from free AA, which become available either from dietary (the end product of digestion) or from metabolic origins as the result of AA biosynthesis within the body. These AA, either circulating via the blood or accumulating within tissues, form pools. The AA concentrations within these pools are based on the equilibrium between gains and losses [18]. Dietary AA are used to build protein for muscle growth, membrane glycoproteins, and enzymes involved in numerous biochemical processes, and act as precursors for the synthesis of DNA/RNA [10]. The AA catabolize in the liver to integrate into protein, which supplies peripheral tissues [13]. Protein turnover refers to the equilibrium between the anabolism and catabolism of protein. The metabolic utilization of AA is equally diverse. Anabolism or protein synthesis facilitates dietary AA to fuse into proteins, or biosynthesize in the body tissues. Catabolism occurs through the breakdown of proteins to build amino groups that produce urea or further protein. In addition, to produces carbon skeleton molecules for glucose production (glucogenesis) or fatty acids (lipogenesis), carbon dioxide, and the release of energy. The role of endocrine hormones in protein metabolism is shown in Table 1.

Nitrogen excretion can be used to determine protein balance through the measurement of nitrogen losses during protein catabolism or recycling [18]. Endogenous or dietary proteins hydrolyze the previous absorption. The tissue proteins of birds are renewed frequently with the liberation of endogenous AA. Furthermore, there are many metabolic reactions converting metabolites into nonessential AA [18].

Recently, AA are applied not only as signaling molecules of the cell and the protein phosphorylation cascade, but also as regulators of gene expression. Moreover, AA are fundamental precursors for hormone synthesis and other nitrogenous elements that have considerable biological significance. Normal levels of AA and their metabolites, such as glutathione, polyamines, taurine, nitric oxide, serotonin, and thyroid hormones are needed for their functions. Nevertheless, elevated levels of AA and their metabolites, such as ammonia, asymmetric dimethylarginine, and homocysteine are considered pathogenic for the body, and lead to oxidative stress, and cause diseases and disorders of the cardiovascular and neurological systems. Therefore, an ideal balance of AA in the feed and bloodstream is crucial for the homeostasis of the body. AA not only have a role as the building blocks of polypeptides and proteins, but also regulate the fundamental metabolic routes that are essential for growth, maintenance, immunity, and reproduction. These functional AA include glutamine, leucine, proline, arginine, cysteine, and tryptophan [10].

## 3. Biosynthesis of Amino Acids

Birds and all vertebrates, dissimilar to plants and many bacteria, are unable to synthesize some AA, so these are termed essential AA and are required for tissue renewal through protein synthesis. Thus, essential AA must be supplemented in the diet. For protein synthesis, all AA are similarly essential owing to the absence of any AA interfere with the anabolic processes. However, nutritionally AA are classified into three groups [18]. Essential AA must be provided by feed and may be classified into two groups. One group is strictly essential because they cannot be synthesized, even from AA metabolic intermediates, such as glucogenic that yield intermediates of glycolysis pathway or ketogenic that yield intermediates of acetyl-CoA or acetoacetate. The transaminases of that group are absent, for instance methionine, lysine, tryptophan, threonine, and phenylalanine. The other group may be insufficiently synthesized from their precursors, for example, glycine, leucine, isoleucine, valine, arginine, histidine, and proline. Semiessential AA may be synthesized from essential AA. Tyrosine and cysteine originate from phenylalanine and methionine, respectively. Cysteine is synthesized from serine (nonessential AA) and methionine (essential AA). Nonessential AA are easily synthesized from intermediary metabolites or similarly nonessential AA: alanine, serine, aspartic and glutamic acids in the former group; and asparagine and glutamine in the latter group [18]. 

## 4. Effect of HS on Protein Metabolism or Turnover

Heat as a stress factor affects protein metabolism during the postabsorptive stage as muscle breakdown and changes in the quantity of lean tissue may occur in different species [21]. In the muscle protein, the RNA/DNA synthesis capacity is reduced by HS [22]. During environmental hyperthermia, muscle tissue catabolism is increased due to increased plasma markers during muscle breakdown. In lactating cows, HS increases plasma urea nitrogen concentration [23]; however, whether this elevation stems from reduced plasma volume, increased protein degradation, or other reasons remains unknown. Thus, blood urea nitrogen (BUN) is used as an indicator of muscle catabolism or breakdown, because tissue degradation results in an increase in BUN [21]. A review [24] reviewed that uric acid excretion is increased in stressed poultry owing to corticosterone -driven gluconeogenesis. Other indicators or measures of protein breakdown (muscle catabolism) include increased plasma creatinine, Nt-methyl histidine, creatine, and creatine kinase (CK) concentrations. An increase in these markers has been detected during heat load in chickens, turkeys [25], cows [26], pigs [27], and humans [28]. The increased level of these parameters indicates enhanced muscle protein catabolism.

Insulin stimulates protein synthesis or accretion. However, during heat-load, increased muscle protein degradation causes the liver to utilize available AA as gluconeogenic substrates from the carbon skeleton through the gluconeogenesis pathway [19]. Under stress conditions, the corticoid hormones (CS, ACTH) suppress the synthesis of tissue proteins and boost proteolysis, as catabolic action is elevated in the blood stream. Glycerol produced from lipid degradation is one of the gluconeogenic substrates and accounts for 20% of the glucose production. Therefore, the other products of protein catabolism are used as substrates for glucose production. First, not only the heart and lung tissue proteins are enhanced by the catabolic transformations of protein composition, but also all tissues except for the nervous system. The muscle tissues (muscle protein) that have the highest body nitrogen content are more sensitive to corticosterone administration, resulting in decreased muscle mass and growth retardation in stressed chickens [29]. A study [30] found that feed deprivation reduced protein synthesis in the liver of starved chickens, as well as plasma albumin and total protein levels. A study [31] indicated that the depletion of plasma free AA, elevated blood uric acid concentration, reduced protein synthesis possibly reflected reduced N retention and more active protein catabolism in broilers challenged by very short-term high temperatures. However, chronic exposure to HS decreased protein digestion, decreased feed digestibility, reduced protein breakdown, reduced protein synthesis in the muscles, and decreased most plasma free AA (especially branched-chain AA and sulfur) [32], whereas the serum levels of glutamic acid, aspartic acid, and phenylalanine increased [33]. It was found that protein synthesis and N deposition were depressed and proteolysis increased during HS [34]. AA catabolism was enhanced under chronic HS [13]; thus, all plasma free AA concentrations decreased, except for glutamic acid, aspartic acid, and phenylalanine. Based on these studies, protein breakdown may increase rapidly in very short-term HS, resulting in a decrease in protein synthesis and an increase in plasma uric acid levels, but then decrease protein breakdown and maintain uric acid levels around normal concentrations as the thermal stress continues. 

Two previous studies, one on chickens and the other on turkeys, found that heat stress reduced uric acid levels in the blood, which could be attributed to a lower level of total protein as a result of hypotonic overhydration [25]. Although no sodium concentration was determined in these studies, water intoxication due to excessive water intake causes overhydration when the amount of water intake exceeds that of water excretion in the kidney. As a result, the sodium level in the blood is diluted, resulting in hyponatremia. As a result, hyponatremia is the most common electrolyte disorder that must be carefully managed [35]. 

There is little knowledge about the renal function of broilers in hot climates, especially in terms of compensating for water and electrolyte loss. During acute heat exposure, there were variable changes in urinary electrolyte excretion in chickens. Reduced glomerular filtration rates (GFR), tubular sodium reabsorption rates, and filtered water amounts may help heat-acclimated birds reduce the metabolic heat load associated with active solute recovery from the glomerular ultrafiltrate. When heat-acclimated birds consume excessive water intake to support evaporative cooling, these changes in kidney function are thought to reduce urinary fluid and solute loss [36]. More research is needed, however, to better explain how various factors may contribute to this evidence.

In addition, a study [37] demonstrated increased uric acid levels in heat stressed chickens. Hence, the application of high-protein diets in HS broilers leads to increased blood plasma uric acid and relieved oxidative stress. Furthermore, the activity of enzymes during AA or protein metabolism under stress conditions has been analyzed. Aspartate aminotransferase (AST) and alanine aminotransferase (ALT) are intracellular enzymes produced in the liver, skeletal muscles, and heart of poultry, and used as indicators of the liver, muscle, and heart damage [38].

The rate of protein accretion is always a constant balance between breakdown (protein-lysis) and synthesis (protein-genesis) [18]. The reduction in protein accretion under conditions of chronic HS is because the rate of protein-genesis is more greatly affected than the rate of proteolysis [33]. Protein synthesis was reduced more in the breast muscles than that in leg muscles; this may be related to higher oxidative metabolism of the leg muscles and increased glycolytic metabolism of the breast muscle [39]. Increasing dietary protein content from 20% to 25% at 32 °C did not affect the rate of protein synthesis but did increase muscle protein deposition, possibly by reducing protein breakdown [33]. The authors of [40] suggested that energy for protein synthesis at the molecular level may be limited at high temperatures; glucose supplementation improves the growth rate at high ambient temperatures. The effects of HS on protein turnover are controversial but may be related to the magnitude and duration of the heat load producing either a detrimental or therapeutic effect. Both HS and pair feeding reduced the muscle mass of rats; however, pair-fed animals had higher protein degradation, leading to a more severe loss of skeletal muscle that might be attributed to protein preservation triggered by heat exposure [41].

## 5. Heat Shock Proteins

Heat stress produces the over-expression of heat shock factors and heat shock proteins (HSP) in bird tissues. HSP regulate multiple molecular pathways in cells in response to stress conditions and change the homeostasis of cells and tissues [1]. HSP affect mediators of inflammation and infection. HSP are molecular chaperones during increased heat, and offer defense. HSP possess mediated responses to endotoxin stimulated synthesis of cytokine, and [42] reviewed that HSP 70 overlap with NFκB transcription, leading to the deactivation of the inflammatory response. Intestinal permeability offers new targets for HS remedy. A study [43] reported that when any living organisms are exposed to HS, the synthesis of most proteins is delayed; however, a group of highly conserved proteins, HSP, is rapidly synthesized. HS causes an increase in HSP synthesis, and are also known as stress proteins [44]. 

A study [45] indicated that HS and subsequent elevated HSP might inhibit muscle mass increase, even with unchanged feed intake. Glutamine seems to have a protective effect on heat-shocked skeletal myotubes by inhibiting protein degradation [46] and this effect might be mediated by HSPs (primarily HSP70 and HSP25/27), independently of glutamine metabolism based on a nonsufficient-metabolizable glutamine analog to mimic the HSP enhancing effect [47]. Additionally, a review [48] reported that increased HSPs defend cells from damage and protect them from apoptosis. HSP 70 is the most common family of HSPs and considered the most conservative, and is plentiful in most living organisms and increases synthesis after cell stress [49]. Glutamine supplementation has been found to increase HSP expression and improve the stress response [47].

## 6. Physiological Mechanism of Stress Regulation in Poultry

According to a review [50], physiological stress regulation mechanisms are classified into three stages: alarm reaction (neurogenic system), resistance or adaptation (endocrine system), and exhaustion. Under HS in fowls, heat generation and metabolizable energy (ME) intake are decreased, which might be owing to reduced thyroid hormones and corticosterone concentration since those endocrine hormones are related to protein turnover acceleration in muscle and thermogenesis [51]. During HS in birds, abnormal pathways occur, including gluconeogenesis, and as protein catabolism increases the efficiency of energy absorption is decreased because of the increased energy retention. Therefore, during periods of stress, it is possible for decreases in growth to be accompanied by increases in body fat deposition [52]. Poultry’s normal body temperature is around 41–42 °C, and the thermoneutral temperature for maximum growth is between 18–21 °C [53]. The environmental temperature, the thermal neutral zone, and the influence of the ambient temperature on heat production and body temperature are shown in Figure 1.

Poultry produce heat through muscular activity and metabolic processes. The optimum or ideal temperature for performance is 19–22 °C in laying hens, and 18–22 °C in broilers [55]. Heat produced in the body is lost through conduction, convection, radiation, evaporation, and fecal excretion. Heat loss falls into two main categories. First, sensible heat loss occurs through convection, conduction, and radiation when hens are in a comfortable environment of 21–25 °C, and show optimum growth rate, egg quality and size, quality of egg shell, egg production, and hatchability. Second, insensible heat loss occurs through panting (evaporative heat loss), and begins when the temperature reaches 26.67 °C [56]. In addition, birds can increase respiration rates up to 10× normal. Additionally, chickens diminish heat by raising and spreading their wings and separating themselves from others. HS has a negative impact on both physiological and behavioral activities. Monitoring these criteria during rearing is critical for identifying HS properties and taking appropriate actions to mitigate the effects of HS while developing high-quality poultry through physiological and management strategies such as heat stress acclimation and poultry housing facilities.

## 7. Hormonal Responses to Stress and the Hormonal Control of Protein Metabolism

Hormone signaling plays a vital role in regulating homeostasis, which includes growth, metabolism, reproduction, and immunity. The overall responses to stress shown in Figure 2. Rapid endocrine responses are mediated by the sympathetic nervous system activation of the adrenal medulla (SA system). However, the long-term effects are due to the activation of the hypothalamic–pituitary–adrenal cortex axis (HPA axis) and the production of glucocorticoids for long time.

The effects of stress on growth performance and reproduction through stress hormonal axis, the reproductive axis, and their interaction are shown in (Figure 3 and Figure 4). The effects of HS on appetite and reproductive hormones are negative. Monitoring appetite and reproductive hormone regulation during rearing are critical for mitigating the negative effects of HS and developing high-quality poultry through hormonal strategies.

## 8. Assessment of Stress

HS has a negative impact on production performance, intestinal health, body temperature, immune responses, appetite hormone regulation, and oxidative properties. It is critical to monitor these criteria during rearing in order to identify HS possessions and take timely action to mitigate the negative effects of high ambient temperature. Stress can be an assessment by three potential methods through behavioral/physiological, endocrine, and metabolic systems measurements. These have been suggested as possible indicators of animal well-being (Table 1) [20]. In addition, neuropeptide Y (NPY) expression is increased in heat-exposed chick brains. NPY has a hypothermic action through the body temperature and heat stress regulation in chicks [58].

## 9. Nutritional Strategies for Preventing HS in Poultry

Nutritional strategies targeted to alleviate and overcome the adverse effects of HS in domestic fowl [59], include preserving feed consumption, electrolytes, water balance, or even by adding vitamins (as ascorbic acid) and minerals [4,5]. Primary strategies in changing the diet formulation of broilers under constant or cycling high-temperature conditions include the suitable use of protein-rich ingredients (AA and crude protein) [12]. It is necessary to ensure the balance of certain AA, especially, the arginine: lysine ratio, and the supplementation synthetic methionine to correct any nutritional shortages [60]. A review [61] found that when protein is the source of energy, the heat increment or specific dynamic action is much greater than when fat or carbohydrate are the sources of energy. Consequently, there are concerns regarding diet-induced heat production related to protein in hyperthermic broilers. Some authors have mentioned the harmful effects of feeding high protein diets [62], leading to the recommendation of a reduced protein diet to control further higher thermogenesis [63]. However, higher dietary crude protein (CP) can compensate reduced AA consumption in stressed broilers, thus it seems to be beneficial in hot conditions, resulting in an improved growth rate [64]. In addition, a review [65] reported that limited protein supplementation decreased water drinking under HS and limits broilers’ performance. Therefore, the AA balance plays a chief role in the scientific conflict regarding the proteins needed for hyperthermic poultry, and it is necessary to determine the AA required for thermoneutrality. Additionally, the protein needed can change gradually after HS exposure, depending on the time exposed. Moreover, a study [66] found lower protein degradation with HS could be normalized with thyroxine supplementation.

Dietary supplementation with one or a mixture of functional AA (glutamine, leucine, proline, arginine, cysteine, and tryptophan) is possibly beneficial. First, for ameliorating or reducing health threats during different periods of the life cycle, such as the metabolic syndrome, fetal growth limitation, weaning-associated wasting syndrome and intestinal dysfunction, neonatal morbidity and mortality, diabetes, obesity, infertility and cardiovascular disease. Second, for improving or optimizing the efficiency of metabolic transformations to boost muscle development, meat and egg quality, and milk production, and reducing adiposity by inhibiting excess fat deposition. Thus, AA has important functions in both health and nutrition [10].

Dietary glutamine supplementation alleviates heat stress, resulting in improved performance and humoral immune response in poultry [67]. In addition, glutamine minimizes the HS effects in heat-stressed chickens in the first weeks of life [68]. Besides it plays several roles in the metabolism and homeostasis of tissues. A study [69] reported that glutamic acid and glutamine supplementation, as a conditionally essential AA in broilers under stress conditions, could be beneficial in improving the growth performance and health. For optimal broiler performance, the use of a high-fat diet (fat is less thermogenic than carbohydrates) with adequate levels of essential AA [70] has been suggested. However, high lysine or Arg:Lys ratio during HS did not reduce the adverse effects of heat stress or even improve the growth rate. Consequently, there is a further challenge to determine the best nutrient during feeding in many fowls during HS [6]. The addition of appropriate feed additives may be beneficial in improving intestinal absorption and minimizing the negative effects of HS. The addition of active substances during incubation is the most recent advancement. By instilling thermotolerance in newly hatched birds, these methods are expected to have an impact on the poultry industry. The physiology, production, and immunological response of broilers under heat stress are all affected by the feeding regimen, which should be tailored to the Ross-308 and Cobb-500 strains [71]. It is necessary to monitor nutritional strategies during nutrition applications in order to prevent HS and produce healthy and comfortable poultry with maintaining feed consumption, dietary adjustments, and appropriate diet formulation. For example, dietary protein-rich ingredients, AA balance, or dietary supplementation with one or a combination of functional AA are all important. Electrolytes, vitamins (such as ascorbic acid), and mineral drinking water supplementation, as well as acid–base balance, are also suggested.

## 10. Effect of HS on Amino Acids

The breakdown of dietary protein results in highly elevated heat generation than that of the catabolism of carbohydrates and fats in poultry under a thermoneutral zone (Table 2).

A study [73] found that feeding broilers more protein than their nutrient requirements did not improve performance at 33 °C. Low protein diets, on the other hand, had a negative impact on broiler performance at high ambient temperatures [64]. A study [74] attributed these effects to lower feed consumption, decreased consumption of AA, and therefore poor body weight gain and feed efficiency. According to a review [75], HS reduced AA levels in the birds including citrulline in chicks’ plasma and leucine in the embryonic brain and liver. As a result, oral L-citrulline increased thermotolerance and decreased body temperature in layer chicks. A review [72] reported that under HS conditions, broilers aged 21 d–49 d should be fed diets containing 90 to 100 percent of the National Research Council (NRC) [76] recommended levels of AA and protein in diets containing 13.4 MJ ME/kg. According to previous studies, nutritionists did not compensate for diminished consumption in the hot ambient temperature by elevating protein and AA levels. Therefore, the final impact on growth relies on optimal or ideal protein quantity. The ideal AA composition for the maintenance or production varies with ambient temperature and among species, which can be attributed to metabolic stress alterations (Table 3).

The optimal or ideal AA for maintenance varies from the ideal AA for production. Birds require higher methionine and cystine, threonine, and fewer leucines than turkeys and pigs, relative to lysine. Some authors reported that greater lysine or Arg:Lys ratios in broiler diets have a beneficial impact, whereas others showed an adverse effect at HT on gain and breast yield [72]. Therefore, dietary AA influenced heat generation [77] and improved broiler performance under high temperatures while decreasing nitrogen excretion by 21% between 28 and 49 days of age [78].

## 11. Future Perspectives and Conclusions

In conclusion, this review discusses the impact and consequences of HS in poultry. In addition, previous work was summarized, and some recommendations for developing high-quality and comfortable poultry through physiological (including HSP regulation), hormonal, and nutritional strategies were provided. Although the influence of HS on protein metabolic conversions in poultry can be concluded from this review, the scientific and medical evidence is inconclusive. Thus, further molecular studies are necessary to determine efficient HS regulation strategies, to better clarify the mechanisms involved in HS tolerance, to understand the HSP family as a useful biomarker for detecting HS. Then, for improved production efficacy in poultry, it is necessary to manage heat stress optimally. Recently, researchers interested in exploring a new generation of genetic tools that are capable of clarifying the molecular pathways associated with the heat stress in chickens, are offering new perspectives for the use of these tools in animal breeding.

## Figures and Tables

**Figure 1 animals-11-01167-f001:**
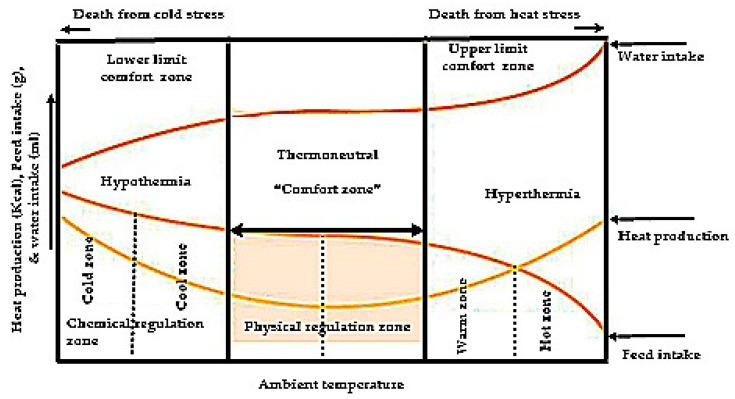
Feed and water intake, and body temperature production related to ambient temperature. Modified after [54].

**Figure 2 animals-11-01167-f002:**
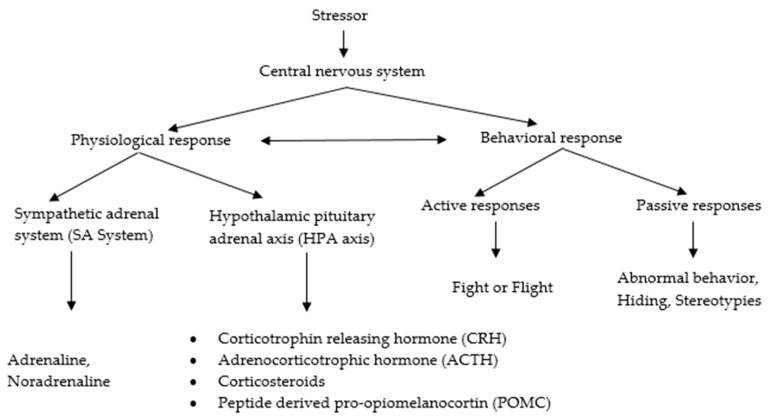
Scheme of overall responses to stress.

**Figure 3 animals-11-01167-f003:**
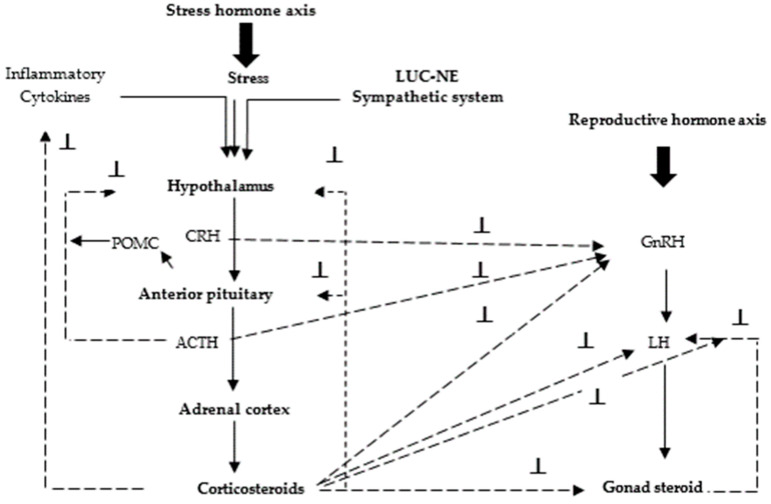
Hypothalamic–pituitary–adrenal axis and its impact on animal reproduction (stress hormone axis include CRH: Corticotropin-releasing hormone; ACTH: Adrenocorticotropic hormone; corticosteroids contain glucocorticoids and mineralocorticoids; POMC: pro-opiomelanocortin; LUC-NE: locus ceruleus neurons which secrete noradrenaline; reproductive hormone axis includes GnRH Gonadotropin-releasing hormone; LH: luteinizing hormone; gonad steroid: testosterone, estradiol, progesterone; **⊥**: inhibition. Modified after [20].

**Figure 4 animals-11-01167-f004:**
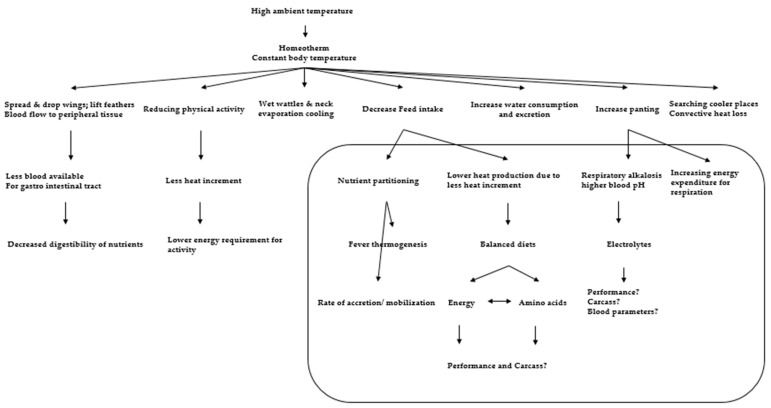
Behavioral and physiological adjustments of chickens at high ambient temperatures and its effects on nutrient intake and utilization. Modified after [57].

**Table 1 animals-11-01167-t001:** Role of endocrine hormones in protein metabolism and summary of potential methods for assessing stress.

Hormones	Protein Synthesis	Proteolysis
Insulin	Stimulated	Inhibited
Glucagon	Inhibited	Stimulated
Epinephrine	Inhibited	Stimulated
Glucocorticoids: ACTH *, CS, and Cortisol	Inhibited	Stimulated (gluconeogenesis)
Thyroid hormones T_4_ and T_3_	Accelerated skeletal muscle protein turnover and heat production under the hot conditions	
Growth hormone	Stimulated	Inhibited
**Potential Methods for Assessing Stress**		
**Behavioral/Physiological**	**Endocrine**	**Metabolic Systems**
Activity/sleep patterns	Catecholamines	Immune function
Posture/stereotypes	ACTH/CRH, glucocorticoids	Disease state
Feed and water intake	Gonadotrophin/sex steroids	Growth performance
Heart rate and blood pressure	Endorphin (β), renin and prolactin	Reproductive performance

* abbreviations: ACTH: adrenocorticotropin, CS: corticosterone, CRH: Corticotropin-releasing hormone, T_3_: triiodothyronine, and T_4_: thyroxine, adapted from [19,20].

**Table 2 animals-11-01167-t002:** The biochemical efficiency of absorbed nutrients for ATP and lipid synthesis; reviewed in [72].

Nutrients	Calorific Value (kJ/g)	ATP Production (%)	Lipid Synthesis (%)
Starch	17.7	68	74
Protein	23.8	58	53
Fatty acids	39.8	66	90

**Table 3 animals-11-01167-t003:** Estimated ideal protein ratio for a starting hen, broiler, and pig, expressed as a lysine needed percentage [70].

Amino Acid	Hen Turkeys	Broiler Chicken	Pigs
Lysine	100	100	100
Methionine + Cystine	59	72	60
Threonine	55	67	65
Valine	76	77	68
Arginine	105	105	NA^1^
Histidine	36	31	32
Isoleucine	69	67	60
Leucine	124	100	111
Phenylalanine + Tyrosine	105	105	95
Tryptophan	16	16	18

NA^1^ = not available.

## Data Availability

Not applicable.

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
