# Peer review of "Protein and Amino Acid Metabolism in Poultry during and after Heat Stress: A Review"

_animals, 2021, doi:10.3390/ani11041167_

Round 1
Reviewer 1 Report
Comments to the Authors of manuscript number: animals-1160653 entitled “Protein and amino acid metabolism in poultry during and after heat stress: a review”.
Authors presented review concerning the role of AA and proteins in heat stress in poultry. The way of thinking is very chaotic. It takes a lot of effort to understand. It should be rephrased.
- L 29 used “affected” is too general. It should be changed
- L 43 – English
- L 48 – coma
- L 52 – not yield. It should be changed
- L 64 – ambient temperature
- L 93 - synthetic processes?
- L 134 – higher animals?
- L 140 – middle metabolites?
- L 177, 178, 238, 323, 349, 368 - By? it is bad manner of writing style
- L 191- it is bad manner of writing style
- L 192 - is it hypotonic over hydration? Was sodium concentration determined?
- L 300 - it is bad manner of writing style
- L320 – elements?
- L 384 0 this paper is review thus it does not indicate but sth can be concluded
Author Response
Thanks for your effort in reviewing our manuscript, the authors have been provided responses requested based on your reviewing comments as following:
Comments and Suggestions Reviewer 1:
Comments to the Authors of manuscript number: animals-1160653 entitled “Protein and amino acid metabolism in poultry during and after heat stress: a review”. The authors presented a review concerning the role of AA and proteins in heat stress in poultry. The way of thinking is very chaotic. It takes a lot of effort to understand. It should be rephrased.
Comment
- L 29 used “affected” is too general. It should be changed.
Response
Thank you very much, the used “affected” are changed as requested and highlighted with yellow color in L 29. “In conclusion, high ambient temperatures significantly reduced dietary AA intake, which in turn reduced protein deposition and growth in broilers”.
Comment
- L 43 – English
Response
Thank you very much for note the phrase corrected as requested and highlighted with yellow color in L 42-44. ”Furthermore, modern commercial broilers are more sensitive to heat stress than previous generations due to their higher performance, growth rate, and feed conversion efficiency”
Comment
- L 48 – coma.
Response
Thank you for your feedback. The comma was removed.
Comment
- L 52 – not yield. It should be changed.
Response
Done as requested and highlighted with yellow color. It phrase (”breast meat yield” was changed to ”breast muscle weight”
Comment
- L 64 – ambient temperature.
Response
As requested, the phrase "environmental temperature" changed to "ambient temperature", and highlighted in yellow.
Comment
- L 93 - synthetic processes?
Response
Thank you for your insightful comment. The phrase "synthetic processes" was changed to "AA biosynthesis" and highlighted in yellow.
Comment
- L 134 – higher animals?
Response
Thank you for your lovely remarks. All vertebrates other than fish are referred to as "higher animals." To avoid readers' confusion, the phrase "higher animals" was changed to "vertebrates animals" and highlighted in yellow.
Comment
- L 140 – middle metabolites?
Response
Thank you for your insightful feedback. To clarify, we replace middle metabolites with the following sentence, which is highlighted in yellow.
“AA metabolic intermediates. Such as glucogenic that yield intermediates of glycolysis pathway. Also, like ketogenic that yield intermediates of acetyl-CoA or acetoacetate”.
Comment
- L 177, 178, 238, 323, 349, 368 - By? it is bad manner of writing style
Response
Thank you incredibly much. All of those sentences have been rewritten in a more professional manner, as requested.
L 177 and L178:
The muscle tissues (muscle protein) that have the highest body nitrogen content are more sensitive, resulting in decreased muscle mass and growth retardation of young organisms; this was observed by [30] in stressed chickens exposed to corticosterone administration. Similar effects were observed by [31] with reduced protein synthesis by the liver of starved chickens and decreased plasma albumin and total protein during feed deprivation. These sentences changed by:
The muscle tissues (muscle protein) that have the highest body nitrogen content are more sensitive to corticosterone administration, resulting in decreased muscle mass and growth retardation in stressed chickens [29]. A study [30] found that feed deprivation reduced protein synthesis in the liver of starved chickens, as well as plasma albumin and total protein levels.
L238,
Three stages of physiological mechanisms of stress regulation have been described by [49]: alarm reaction (neurogenic system), resistance or adaptation (endocrine system), and exhaustion. This replaced by
“According to a review [49], physiological stress regulation mechanisms are classified into three stages: alarm reaction (neurogenic system), resistance or adaptation (endocrine system), and exhaustion.”
L323,
The [57] reported that dietary protein has been considered because if shows higher heat production released from its metabolism higher than fats or carbohydrates. This sentence changed by:
A review [57] found that when protein is the source of energy, the heat increment or specific dynamic action is much greater than when fat or carbohydrate are the sources of energy.
L349,
Also, [65] reported that the dietary mixture of L-glutamine and L-glutamic supplementation could be beneficial in improving heat tolerance in broilers. This sentence changed by:
A study [64] reported that glutamic acid and glutamine supplementation as a conditionally essential AA in broilers under stress conditions such as a hot tropical condition could be beneficial in improving the growth performance.
L368,
During HS, birds in the finisher period must be provided diets comprising approximately 90%– 100% of AA and protein, and according to the [68] foods should have a 13.4 MJ/kg metabolic energy [66].
This sentence changed by:
A review [66] reviewed that under HS conditions, broilers aged 21-49d should be fed diets containing 90 to 100 percent of the NRC (1994) [68] recommended levels of AA and protein in diets containing 13.4 MJ ME/kg.
Comment
- L 191- it is bad manner of writing style
Response
Thank you, the L 191. Changed from “In the studies of [36] and [37] observed that HS decreased blood uric acid levels in chickens and turkeys because the total protein was reduced owing to hypotonic over hydration induced by excessive water intake ”. To the following:
Two previous studies, one on chickens and the other on turkeys, found that heat stress reduced uric acid levels in the blood, which could be attributed to a lower level of total protein as a result of hypotonic overhydration [25].
Comment
- L 192 - is it hypotonic over hydration? Was sodium concentration determined?
Response
Thank you for your insightful comment. The answer to this valuable comment added to the text and highlighted with yellow color as follows:
“Although no sodium concentration was determined in these studies, water intoxication due to excessive water intake leading to overhydration and when the amount of water intake exceeds that of water excretion in the kidney. As a result, the sodium level in the blood is diluted, resulting in hyponatremia. As a result, hyponatremia is the most common electrolyte disorder that must be carefully managed [36].
There is little knowledge about the renal function of broilers in hot climates, especially in terms of compensating for water and electrolyte loss. During acute heat exposure, there were variable changes in urinary electrolyte excretion in chickens. Reduced glomerular filtration rates (GFR), tubular sodium reabsorption rates, and filtered water amounts may help heat-acclimated birds reduce the metabolic heat load associated with active solute recovery from the glomerular ultrafiltrate. When heat-acclimated birds consume excessive water intake to support evaporative cooling, these changes in kidney function are thought to reduce urinary fluid and solute loss [37]. More research is needed, however, to better explain how various factors may contribute to this evidence.”
Comment
- L 300 - it is bad manner of writing style
Response
Done as requested.
Comment
- L320 – elements?
Response
Thank you. "Ingredients" are referred to like elements in this context. As a result, the phrase "protein-rich elements" was changed to "protein-rich ingredients" and highlighted in yellow to better explain it.
Comment
- L 384 0 this paper is review thus it does not indicate but sth can be concluded
Response
Thank you for your insightful comments. At this point, the authors change their writing style to say, “Although the influence of HS on protein metabolic conversions in poultry can be concluded from this review, the scientific and medical evidence is inconclusive”.
Reviewer 2 Report
In this review, authors described the biosynthesis of amino acids and/or protein metabolism under normal conditions and heat stress condition. Summarized the role of endocrine hormones in protein metabolism, and the nutritional strategies for the prevention of heat stress in poultry. They only simple accumulated the studies, not summarized them. The review was not a good job for summary the previous works and provide some advises to develop high grade poultry. There are several disadvantages as follow:
The authors will need to seek assistance from an English language editor to revise, there are too many long sentences to read.
This review is too exhaustive to point. All of effect of the HS was written, but it only to write, not review. If Author can summary some points, as physiological (including HSP regulation), hormonal, and nutritional strategies, to discuss the effect of HS in poultry, I think this is valuable. The current version is not suitable for scientific paper.
L50, some words as 'heat loss' are better than 'heat dissipation' in this manuscript, authors should wording them.
L84-149, I stronger recommend delete them or rewrite. Authors should review around the heat stress. Now, the part is describing the biosynthesis for AA or protein.
I recommend authors re-draw the figures. As Figure 3, the thermal neutral zone is too widen. And it is error food intake increases with temperature decreases, it will a limit temperature as Figure 3B.
L40: I suggest here add 'a review' before [2]
L70: I suggest here add 'a study before [13]
L42: their? What's it?
Reference: Please conform the format of the reference according to the Journal. In the main text, I don’t think the references as L191. I suggested authors add the first author of the reference at in the main text. Please check other sites.
And the species names should be italic.
Author Response
Thanks for your effort in reviewing our manuscript, the authors have been provided responses requested based on your reviewing comments as following:
Reviewer 2: Comments and Suggestions
Comment
In this review, authors described the biosynthesis of amino acids and/or protein metabolism under normal conditions and heat stress condition. Summarized the role of endocrine hormones in protein metabolism, and the nutritional strategies for the prevention of heat stress in poultry. They only simple accumulated the studies, not summarized them. The review was not a good job for summary the previous works and provide some advises to develop high-grade poultry. There are several disadvantages as follow:
The authors will need to seek assistance from an English language editor to revise; there are too many long sentences to read.
Response
Thank you incredibly much. However, before submission, the King Saud University Researchers Support & Services Unit edited this review for English language, grammar, punctuation, and spelling. In addition, we were able to shorten long sentences with the help of an English language expert, as requested.
Comment
This review is too exhaustive to point. All of effect of the HS was written, but it only to write, not review. If Author can summary some points, as physiological (including HSP regulation), hormonal, and nutritional strategies, to discuss the effect of HS in poultry, I think this is valuable. The current version is not suitable for scientific paper.
Response
Thank you so much for your wise counsel. As requested and highlighted in yellow, the authors summarized previous works and provided some recommendations for developing high-quality poultry through physiological (including HSP regulation), hormonal, and nutritional strategies.
Such as:
The addition of appropriate feed additives may be beneficial in improving intestinal absorption and minimizing the negative effects of HS. The addition of active substances during incubation is the most recent advancement. By instilling thermotolerance in newly hatched birds, these methods are expected to have an impact on the poultry industry.
HS has a negative impact on production performance, intestinal health, body temperature, immune responses, appetite hormone regulation, and oxidative properties. It is critical to monitor these criteria during rearing in order to identify HS possessions and take timely action to mitigate the negative effects of high ambient temperature.
The effects of HS on appetite and reproductive hormones are negative. Monitoring appetite and reproductive hormone regulation during rearing are critical for mitigating the negative effects of HS and developing high-quality poultry through hormonal strategies.
HS has a negative impact on both physiological and behavioral activities. Monitoring these criteria during rearing is critical for identifying HS property and taking appropriate actions to mitigate the effects of HS while developing high-quality poultry through physiological and management strategies such as heat stress acclimation and Poultry housing facilities.
It is necessary to monitor nutritional strategies during nutrition applications in order to prevent HS and produce healthy and comfortable poultry. Maintaining feed consumption, dietary adjustments, and appropriate diet formulation. For example, include the use of protein-rich ingredients, AA balance, or dietary supplementation with one or a combination of functional AA. Electrolytes, vitamins (such as ascorbic acid), and mineral drinking water supplementation, as well as acid-base balance, are also suggested.
Comment
L50, some words as 'heat loss' are better than 'heat dissipation' in this manuscript, authors should wording them.
Response
Thank you very much for your note. The phrase "'heat dissipation" was changed to " heat loss" and highlighted in yellow in L25 and L50.
Comment
L84-149, I stronger recommend delete them or rewrite. Authors should review around the heat stress. Now, the part is describing the biosynthesis for AA or protein.
Response
Thank you very much. The authors deleted some paragraphs in those sections that deviated from protein and amino acid metabolism under normal and HS conditions. In addition, they rewrote sentences that were written in an incorrect manner.
Comment
I recommend authors re-draw the figures. As Figure 3, the thermal neutral zone is too widen. And it is error food intake increases with temperature decreases, it will a limit temperature as Figure 3B.
Response
Thanks for drawing our attention to re-draw the figure 3 and avoid that error. Done as requested.
Comment
L40: I suggest here add 'a review' before [2]
Response
Done as requested and highlighted with yellow color. We add 'a review' before [2].
Comment
L70: I suggest here add 'a study before [13]
Response
Done as requested and highlighted with yellow color. We add 'a study' before [13].
Comment
L42: their? What's it?
Response
Thank you so much, "Their" is a pronoun returning on commercial broilers. As a result, we rewrote the sentence as follows. “Furthermore, modern commercial broilers are more sensitive to heat stress than previous generations due to their higher performance, growth rate, and feed conversion efficiency”.
Comment
Reference: Please confirm the format of the reference according to the Journal. In the main text, I don’t think the references as L191. I suggested authors add the first author of the reference in the main text. Please check other sites. And the species names should be italic
Response
Thank you so much. We double-checked the references and confirmed their format in accordance with the Journal reference style. In addition, the authors rewrote and checked references L191.
Finally, thank you for your insightful comments and valuable suggestions, which will significantly improve our review's quality and scientific background.
Reviewer 3 Report
Dear Authors,
I kindly accepted the invitation of review on the paper: “Protein and amino acid metabolism in poultry during and after heat stress: a review”.
The article reviews the effects of the environmental heat stress on protein and amino acid metabolism, physiology, welfare and growth performance of broiler chickens. It explores the potential strategies and interventions to mitigate the heat stress. Manuscript is well written and information is presented in logical order. Authors reviewed a large number of recent and older peer-reviewed publications, thus it ensures a good quality and variety of sourced research. Authors often present the basic knowledge; however, I believe it was used to maintain the flow of information.
Overall, it is informative and quite pleasant to read.
Specific comments:
Line 87- please use the AA abbreviation earlier in the text, and use AA rather than AAs.
Author Response
Thank you for your effort in reviewing our manuscript, the authors have changed the manuscript according to your suggestions and replied to all the valuable comments as follows:
Review 3: Comments and Suggestions for Authors
Dear Authors,
I kindly accepted the invitation of review on the paper: “Protein and amino acid metabolism in poultry during and after heat stress: a review”.
The article reviews the effects of the environmental heat stress on protein and amino acid metabolism, physiology, welfare and growth performance of broiler chickens. It explores the potential strategies and interventions to mitigate the heat stress. Manuscript is well written and information is presented in logical order. Authors reviewed a large number of recent and older peer-reviewed publications, thus it ensures a good quality and variety of sourced research. Authors often present the basic knowledge; however, I believe it was used to maintain the flow of information.
Overall, it is informative and quite pleasant to read.
Specific comments:
Line 87- please use the AA abbreviation earlier in the text, and use AA rather than AAs.
Response
Thank you very much for your insightful recommendation and counsel. The AA abbreviation is now used earlier in the text, with AA rather than AAs, and the entire text is highlighted in yellow.
Thanks so much
Finally, so many thanks to Editor and Reviewers for the valuable comments and suggestions that will significantly improve a review.
Round 2
Reviewer 1 Report
I have not additional comments
Author Response
Comments and Suggestions Reviewer 1:
Comment
I have not additional comments.
Response
Thank you for your approval of our revisions based on the “Review Report (Round 1). In addition, thank you for your that’s insightful comments and valuable suggestions, which will significantly improve the quality and scientific background of our review.
Reviewer 2 Report
In the new version, I cant find the change as "
Comment
L40: I suggest here add 'a review' before [2]
Response
Done as requested and highlighted with yellow color. We add 'a review' before [2]."
Plaese submit the changed version to me.
Then I will review it.
Author Response
Reviewer 2: Comments and Suggestions
In the new version, I cant find the change as "
Comment
L40: I suggest here add 'a review' before [2]
Response
Done as requested and highlighted with yellow color. We add 'a review' before [2]."
Plaese submit the changed version to me.
Then I will review it.
Response
Oh dear Dr. Reviewer 2, we forgot to include it in the text, despite including it in the previous response.
We've placed 'a review' before [2], as well as before any other sites that required it. As a result, we added 'a review' and highlighted it in orange in L40, L70, and L333.
In addition, we insert 'a study' and highlight it in orange in L176, L207, L237, L241, and L338.
Thank you for your previous and new insightful comments and valuable suggestions, which will greatly improve the quality and scientific foundation of our review.
Round 3
Reviewer 2 Report
The version is better. I recommend "Accept after minor revision"
provide the figure with high quality. check references list, as L438, L445. it is not only limit those. rewrite L375-376.Author Response
Comment
The version is better. I recommend "Accept after minor revision"
provide the figure with high quality. check references list, as L438, L445. it is not only limit those. rewrite L375-376.
Response
The author was redraws figures with high quality based on your comments and suggestions (Round 3). Furthermore, we checked all references as L438 and L445 and highlighted them in Turquoise. In addition, we rewrite L375-376.
Thank you for your previous and new insightful comments and valuable suggestions, which will greatly improve the quality and scientific foundation of our review.